# The Correlation between Morpho-Dynamic Contrast-Enhanced Mammography (CEM) Features and Prognostic Factors in Breast Cancer: A Single-Center Retrospective Analysis

**DOI:** 10.3390/cancers16050870

**Published:** 2024-02-22

**Authors:** Claudia Lucia Piccolo, Ilenia Celli, Claudio Bandini, Manuela Tommasiello, Matteo Sammarra, Lorenzo Faggioni, Dania Cioni, Bruno Beomonte Zobel, Emanuele Neri

**Affiliations:** 1Department of Radiology, Fondazione Policlinico Universitario Campus Bio-Medico, 00128 Rome, Italy; c.piccolo@policlinicocampus.it (C.L.P.); ilenia.celli.fw@fbf-isola.it (I.C.); m.tommasiello@policlinicocampus.it (M.T.); m.sammarra@policlinicocampus.it (M.S.); 2Department of Translational Research, Academic Radiology, University of Pisa, 56126 Pisa, Italy; lorenzo.faggioni@unipi.it (L.F.); dania.cioni@unipi.it (D.C.); emanuele.neri@unipi.it (E.N.); 3Department of Medicine and Surgery, Università Campus Bio-Medico di Roma, Via Alvaro del Portillo, 00128 Roma, Italy; b.zobel@policlinicocampus.it; 4Operative Research Unit of Diagnostic Imaging, Fondazione Policlinico Universitario Campus Bio-Medico, 00128 Roma, Italy

**Keywords:** contrast-enhanced mammography, breast cancer, magnetic resonance imaging, contrast media

## Abstract

**Simple Summary:**

Breast cancer remains a significant global health concern, demanding effective detection methods. This study explores Contrast-Enhanced Mammography (CEM), an alternative to digital mammography, for its potential in correlating with breast cancer prognostic factors. Analyzing 182 lesions in 114 women, the research identifies associations between CEM features (such as spiculated margins and irregular shape) and key prognostic factors, including tumor grade and molecular markers. Notable findings suggest that certain CEM characteristics may predict aspects of breast cancer behavior, providing valuable insights for personalized treatment decisions. While acknowledging study limitations, the results highlight the promise of CEM in contributing to our understanding of breast cancer biology and guiding clinical approaches. Further research is essential to validate and expand upon these findings.

**Abstract:**

Breast cancer, a major contributor to female mortality globally, presents challenges in detection, prompting exploration beyond digital mammography. Contrast-Enhanced Mammography (CEM), integrating morphological and functional information, emerges as a promising alternative, offering advantages in cost-effectiveness and reduced anxiety compared to MRI. This study investigates CEM’s correlation with breast cancer prognostic factors, encompassing histology, grade, and molecular markers. In a retrospective analysis involving 114 women, CEM revealed diverse lesion characteristics. Statistical analyses identified correlations between specific CEM features, such as spiculated margins and irregular shape, and prognostic factors like tumor grade and molecular markers. Notably, spiculated margins predicted lower grade and HER2 status, while irregular shape correlated with PgR and Ki-67 status. The study emphasizes CEM’s potential in predicting breast cancer prognosis, shedding light on tumor behavior. Despite the limitations, including sample size and single-observer analysis, the findings advocate for CEM’s role in stratifying breast cancers based on biological characteristics. CEM features, particularly spiculated margins, irregular shape, and enhancement dynamics, may serve as valuable indicators for personalized treatment decisions. Further research is crucial to validate these correlations and enhance CEM’s clinical utility in breast cancer assessment.

## 1. Introduction

Breast cancer is the most frequently diagnosed cancer in women, with approximately one in three malignancies being of breast origin, and is the leading cause of death among women worldwide [1]. Digital mammography is the gold standard imaging test in breast cancer screening. It demonstrated to have a significant impact on survival and a reduction in mortality, estimated at 18% in a meta-analysis performed on eight randomized studies [2]. However, several studies showed that from 20 to 30% of breast cancers are undetectable by means of digital mammography [3,4,5,6]. To date, Magnetic Resonance Imaging (MRI) is the gold standard technique to provide functional information about tumor neoangiogenesis, improving the detection and characterization of malignant lesions. MRI is characterized by high sensitivity and offers advantages, such as a reduced radiation exposure for patients, the use of contrast agents with fewer side effects compared to iodine-based ones, and the simultaneous assessment of axillary lymph node metastases during the examination, but it is burdened by a lot of limitations, including high costs, long acquisition times and limited availability. Contrast-Enhanced Mammography (CEM) is a relatively recent technique that has proven to overcome the limitations of digital mammography [6]. This technique combines the morphological information of digital mammography with the functional information obtained from the administration of iodinated contrast material injected intravenously [7,8,9], allowing for the evaluation of tumor neoangiogenesis, similar to MRI. This method has shown diagnostic performance comparable to MRI (sensitivity 96–100%) [7,8,9,10] and comes with additional advantages, like lower costs [11], shorter execution times, greater tolerability and less anxiety in patients [12,13,14], even though it exposes the patient to ionizing radiation and requires the use of iodine-based contrast agents. The main indications for CEM are: detection and pre-operatory evaluation of breast cancer [15,16,17,18], problem-solving [19], screening patients with high-risk symptoms [20,21] and post-treatment staging of tumors [22,23,24,25]. This study aims to evaluate the morphological and functional features of breast cancer on CEM examination (shape, margins, dimensions and enhancement of the target lesion, according to ACR BI-RADS Mammography V edition) [26,27,28,29] that can predict the prognostic factors such as the histological type, histological grade, estrogen receptor (ER), progesterone receptor (PgR), Ki67, human epidermal growth factor receptor 2 (HER2) status and node invasion and, thus, the outcome of the disease.

## 2. Materials and Methods

### 2.1. Study Population

This retrospective and single-center study was performed in accordance with the Declaration of Helsinki and approved by the ethics committee of our Hospital (study protocol: 51.23 OSS).

From August 2021 to May 2023, 114 women with a histological diagnosis of breast cancer underwent contrast-enhanced mammography (CEM) at the Breast Unit of the Fondazione Policlinico Universitario Campus Bio-Medico in Rome, Italy.

The inclusion criteria were: a suspicious breast lesion (BI-RADS 4 or 5) found on conventional imaging (mammography/tomosynthesis or ultrasound examination), patients older than 18 years of age and patient able to perform CEM examination after signing the informed consent.

Exclusion criteria were: pregnancy, iodinated contrast material allergy, renal failure and breast prostheses.

Before the examination, renal function and coagulation parameters were evaluated and written informed consent was obtained.

### 2.2. Contrast-Enhanced Mammography (CEM)

A digital mammography unit (Senographe Pristina, GE Healthcare system) was used to perform CEM examinations. In our center, CEM consists of the acquisition of the low energy (25–29 kVp) and high energy (45–49 kVp) images with the Dual Energy technique after two minutes from the administration of intravenous iodinated contrast medium (Omnipaque 350 mg/mL). Before the examination, a contrast agent is administrated through an antecubital vein, preferring the contralateral arm of the target lesion. The contrast dose was usually 1.5 mL/kg body weight, at a rate of 2.5 mL/s, followed by 20 mL of saline flush automatically injected at 3 mL/s. The two images are processed using subtraction algorithms with the production of a combined mammographic image to enable the possibility of analyzing the dynamics of enhancement of a suspected lesion, in a similar way to MRI. To obtain the recombined image the low and high-energy images are processed by means of a three steps subtraction. Specifically, at first the images are log-transformed using the natural logarithms (1) and (2):(1)$$\ln(I_{d}^{LE})=\ln(I_{0}^{LE}\cdot e^{-\left(\mu_{t}^{LE}\cdot t+\mu_{1}^{LE}\cdot T\right)})=\ln(I_{0}^{LE})-\left(\mu_{t}^{LE}\cdot t+\mu_{1}^{LE}\cdot T\right)$$
(2)$$\ln(I_{d}^{HE})=\ln(I_{0}^{HE}\cdot e^{-\left(\mu_{t}^{HE}\cdot t+\mu_{t}^{HE}\cdot T\right)})=\ln(I_{0}^{HE})-\left(\mu_{t}^{HE}\cdot t+\mu_{1}^{HE}\cdot T\right)$$

Then, the log-transformed low-energy image is multiplied by a weighting factor w, which depends on the low- and high-energy attenuation coefficients of normal breast tissue, which depends on the used spectra.

The w has a value chosen to eliminate the attenuation of the normal tissue in the final recombined image. The subtraction can be written as (3):(3)$$\ln(I_{d}^{HE})-w\cdot \ln(I_{d}^{LE})=\left[\ln(I_{0}^{HE})-\left(\mu_{t}^{HE}\cdot t+\mu_{1}^{HE}\cdot T\right)\right]-w\cdot \left[\ln(I_{0}^{HE})-\left(\mu_{t}^{HE}\cdot t+\mu_{1}^{HE}\cdot T\right)\right]$$

In this function, there is one term that contains the thickness of the normal tissue t and one term containing the thickness of the lesion T. The natural logarithm of the incoming intensities is a constant value, indicated by C in the final equation.

In the final step, the weighted log-transformed low-energy image is subtracted from the log-transformed high-energy image, obtaining the so-called “recombined image”, which is an image in which the areas of iodine accumulation or “enhancement” are clearly visible. In this process, it is necessary to choose the weighting factor w so that the term containing the tissue thickness t becomes zero. The used low- and high-energy X-ray radiation define the value of m (low energy) and m (high energy), and thus the numerical value of the weighting factor w.

The resulting equation shows only a dependency on the lesion thickness (4):(4)$$\ln(I_{d}^{HE})-w\cdot \ln(I_{d}^{LE})=C-\left[\mu_{1}^{HE}-w\cdot \mu_{1}^{LE}\right]\cdot T$$

In the second step, it is fundamental to choose the weighting factor w so that the normal tissue is eliminated, while the iodine contrast agent not. The weighting factor is decided based on the k-edge of iodine; in fact, for the breast tissue, the attenuation coefficient gradually decreases by increasing the photon energy, while for the iodine, there is a significant increase at the k-edge. These differences in attenuation coefficients between low- and high-energy for tissue and iodine allow to enhance the iodine signal, resulting in an image that is dominated by the iodine signal only.

To summarize, the X-ray radiation is attenuated by the lesion and the normal tissue. The level of attenuation is defined by the attenuation coefficient of the material multiplied by the thickness: (m m t L Å~t + LE Å~T) and (m m t Hel Å~t + HE Å~T) for the low and high-energy image, respectively. According to the Beer–Lambert law, the intensity of the incoming X-rays is attenuated exponentially when passing through the tissue and the lesion. Each voxel in both the low- and high-energy image contains information of the lesion and the normal tissue. Because two images are acquired with different energies, the iodine attenuation from the lesion can be unraveled by manipulating these two equations.

### 2.3. Imaging Examination

The exam was performed as follows: Craniocaudal (CC) of the contralateral side, Craniocaudal (CC) of the target lesion, Medio-lateral (MLO) of the target lesion and Medio-lateral (MLO) of the contralateral side. If an enhancement was observed on the suspicious side, an additional image was taken after eight minutes in order to assess the enhancement kinetics and establish the probability of malignancy.

The images were reviewed by a radiologist with 10 years of breast imaging experience and five years of CEM evaluation, who viewed the exams on a high-resolution workstation.

The CEM images were analyzed following the American College of Radiology (ACR) BI-RADS CEM lexicon [27].

We classified the suspicious lesions as “mass” and “non-mass” enhancement. For each lesion, we assessed the shape, the margins, the dimensions and the enhancement. The shape of a mass was described as round, oval, or irregular. The margins of a mass were described as circumscribed or not circumscribed. The dimensions of the index lesions were measured by reporting the maximum diameter in the early scans and looking for the wash-out in the late ones, if any. The cut-off of the dimensions of each identified lesion was considered 2 cm. The enhancement pattern was evaluated as homogeneous, heterogeneous, or rim enhancement. In the case where an enhancement was not observed, the index lesion was measured as 0 mm. Larger focus extension has been reported in cases of multicentric and multifocal invasive tumors. We analyzed the enhancement kinetics looking for the wash-out or wash-in in the late images.

### 2.4. Histological Examination

Suspicious lesions classified as BI-RADS 4 or 5 on conventional imaging were biopsied using core needle biopsy (CNB) or vacuum-assisted breast biopsy (VAAB). Histological examination was performed by a pathologist with over 25 years of experience in breast disease, according to World Health Organization (WHO) guidelines. The histology, type and grade of the tumor and receptor structure (ER, PgR and HER2), the Ki67 proliferation index and nodes involvement were analyzed. HER2 was considered positive (value 1) with a value of 3+, while HER2 value 1 results were considered negative (value 0). Specimens yielding an equivocal immunohistochemical result (2+) underwent an analysis by fluorescent in situ hybridization (FISH). In case of amplification, a value of 1 (positive) was assigned to HER2; in case of unamplified FISH, the value assigned was 0. The cut-off for Ki67 positivity was 14%.

### 2.5. Statistical Analysis

A statistical analysis was conducted using IBM SPSS version 27.0. Categorical variables were presented as frequency (N) and percentage (%), while ordinal and numerical variables were described using median and absolute range values. The comparison of categorical variables was carried out using either the χ^2^ test or Fisher’s exact test. Logistic regression was employed to assess the association between Contrast Enhanced Mammography (CEM) features and prognostic factors. The logistic regression model can be expressed as follows (5):(5)$$p=\frac{1}{1+e^{-\left(\beta_{0}+\beta_{1}x_{1}+\beta_{2}x_{2}+\cdots +\beta_{n}x_{n}\right)}}$$
where *p* represents the probability of the outcome (e.g., presence of a prognostic factor); *e* is the base of the natural logarithm and β are the regression coefficients associated with each predictor variable (*x*_1_, *x*_2_… *x_n_*).

The logistic function $\frac{1}{1+e^{-z}}$ transforms the linear combination of predictor variables into a probability value between 0 and 1.

The logistic regression analysis results were reported with regression coefficients (β) along with their standard errors and *p*-values, as well as odds ratios (OR) with 95% confidence intervals (CI). A significance level of *p* < 0.05 was used to determine statistical significance.

## 3. Results

### 3.1. Imaging Analysis

Among the 114 patients enrolled in the study, 102 patients had unilateral tumor (50 cases were on the right breast and 52 on the left breast), 10 patients had bilateral lesions, and two patients had synchronous unilateral breast tumors (i.e., two different lesions in the same breast). The age ranged between 31 and 85 years, with a median of 56 years. A total of 182 lesions were analyzed: 102 (56%) presented as mass lesions, 76 (42%) as mass and non-mass enhancement lesions, and four (2%) were lesions without enhancement (Table 1). At CEM examination there were 52 (51%) oval or round shape masses and 50 (49%) irregular shape lesions. In terms of margins, there were 18 (18%) circumscribed margin lesions and 84 (82%) non-circumscribed (irregular or spiculated) margin lesions. A total of 48 (47%) lesions revealed a diameter of less than 2 cm, while the diameter was greater than or equal to 2 cm for 54 (53%) cases. There were 28 (27%) lesions characterized by homogeneous enhancement, 70 (69%) cases by heterogeneous enhancement and four (4%) cases by rim enhancement. In total, 28 (27%) lesions showed a wash-in enhancement in the late images, while 74 (73%) lesions showed a wash-out enhancement (Table 2 and Figure 1).

The distribution and the internal enhancement on non-mass lesions were analyzed: in 52 (68%) cases, there was a linear non-mass enhancement, and in 24 (32%) cases, there was a segmental non-mass enhancement. In total, 20 (26%) cases were characterized by homogeneous internal enhancement lesions, 44 (58%) heterogeneous internal enhancement lesions, and 12 (16%) clumped enhancement lesions.

### 3.2. Histopathological Analysis

Among 182 biopsies examined, histological grade tumor G1 was found in 40 (22%) cases, while G2-G3 was found in 142 (78%) cases. The histology indicated that node invasion was present in 40 (22%) cases, while 142 (78%) did not show it. In total, 16 lesions (8%) were ER negative and 166 (92%) ER positive; 32 lesions (18%) were PR negative and 150 were (82%) PR positive; 32 lesions (18%) were Ki67 negative and 150 (82%) were Ki67 positive; 140 lesions (77%) were HER2 1 or 2 and 42 (23%) were HER2 3 (Table 3). The distribution of histological types of the 182 lesions was: 80 (44%) cases of invasive breast cancer (68 invasive ductal cancer, CDI, and 12 invasive lobular cancer, CLI), and 102 (56%) cases of invasive breast cancer (ductal or lobular) + carcinoma in situ (CDI or CLI + Cis). No cases of carcinoma in situ alone (Cis) were found (Table 4).

### 3.3. Statistical Analysis

By univariate analysis the shape of a mass was significantly correlated with the ER (*p* = 0.006), Ki-67 expression status (*p* = 0.005), mass dimensions (*p* < 0.001) and with nodes involvement (*p* = 0.03). The margins of a mass were significantly correlated with the histological grade (*p* = 0.02), dimensions (*p* < 0.001), expression of PR (*p* = 0.01), and Her 2 status (*p* = 0.002). The dimensions measured on CEM images were significantly associated with the histological grade (*p* = 0.01), the ER (*p* < 0.001), and PgR expression status (*p* = 0.009) and Ki-67 expression status (0.02). The internal enhancement pattern was significantly associated with lesion dimensions (*p* < 0.001), expression of ER (*p* < 0.001), expression of PR (*p* < 0.001), Ki-67 status (*p* = 0.04) and nodes involvement (*p* < 0.05). The late phase pattern was significantly correlated with nodes involvement (*p* = 0.01), with the ER expression status (*p* = 0.005), and with the histological subtype (*p* = 0.004) (Table 5 and Table 6). A correlation was not found between CEM breast lesion presentations (mass/non mass enhancement) and prognostic factors. Three parameters found to be significant by univariate analysis were selected for logistic regression analysis. Regression analysis data are summarized in Table 7; in particular, spiculated margins were significant, independent predictor factors of nodes involvement (*p* = 0.02), a low histologic grade (*p* = 0.01) and Her 2 lower expression (*p* < 0.001). The irregular shape lesion was a significant independent predictor factor of a low risk of nodes involvement (*p* = 0.038), a lower Prg Expression status (*p* = 0.004) and higher Ki-67 status (*p* = 0.03). The wash out in the late phase was a significant independent predictor factor of a lower Er and PgR expression status (*p* = 0.02).

## 4. Discussion

Breast cancer prognosis relies on several morphologic features (among which axillary nodes status is the single most relevant prognostic factor) and on molecular ones. In particular, the evaluation of Er and PgR status is essential to select patients who can benefit from hormonal therapy, HER2 receptor status helps to select patients sensitive to Trastuzumab, whereas Ki-67 evaluation demonstrated to be an independent prognostic factor for node-negative patients. In this way prognostic factors allow to estimate patient risk of developing a micro-metastatic diseases and to select patients potentially suitable for adjuvant therapy. The development of metastasis is closely related to angiogenesis and contrast leakage from vessels into the tumor interstitium. The gold standard in breast tumor staging is MRI, which takes advantage of the formation of new vessels to recognize neoplastic cells [30,31]; nevertheless, MRI is characterized by some drawbacks, such as the low availability, the long acquisition times, and the low specificity, leading to further examinations in some specific cases, causing a lengthening in the decisional process. In the last few years, Contrast Enhanced mammography demonstrated encouraging results in breast cancer detection and staging, with a sensitivity higher than digital mammography (DM) and similar to that of MRI [32,33,34]. Unlike MRI, CEM seems to improve cancer detection without decreasing specificity, since it provides higher contrast and better lesion evaluation, recognizing more multifocal breast cancers in a better way than mammography alone or combined with ultrasonography [35,36,37,38]. Despite the rich literature on CEM imaging, only a few studies have focused on the relationship between CEM features and prognostic factors in breast cancer, in particular by adding information coming from the late phase [38,39,40]. In our study, spiculated margins were able to predict a lower histological grade, a lower HER-2 status and a lower risk of nodes involvement; this result is in line with the studies performed by means of MRI, according to which high-grade tumors manifest at imaging as well-defined masses because of their rapid growth, while intermediate and low-grade tumors, because of their low cellularity and rich collagen matrix, involve a desmoplastic host reaction and appear as spiculations [39,40,41]. An irregular shape (i.e., architectural distortion) demonstrated to predict a lower Prg Expression status (a sign of poor prognosis) and higher Ki-67 status, indicating a lesion with a high cell rate proliferation and a poor outcome, but it was a protective factor for the risk of node involvement. The qualitative analysis of lesion enhancement demonstrated that the presence of wash-out was a strong predictor of lesions with lower Er and PgR expression status, correlated to a poor prognosis; this result can be explained by the neonagiogenesis phenomenon, which occurs in neoplastic lesions, characterized by higher vascular permeability, because of a higher VEGF expression, which demonstrated to correlate with higher histologic grade, overexpression of HER2 and Ki-67 proliferation [42,43]. Our study has some limitations, such as the small sample size, which reduces the statistical power of the analysis performed; the images were interpreted by one observer only, so interobserver variability could not be analyzed; and the impossibility of comparing CEM results with MRI ones, in order to evaluate which one better correlated with prognostic factors.

## 5. Conclusions

The morphologic CEM features of breast cancer indicate that spiculated margins may predict a favorable prognosis, while the irregular shape and the wash-out in the late phase an unfavorable prognosis. These features may help the radiologist to select subgroups of breast cancers at imaging with different biological behavior. More studies with larger a cohort are needed to confirm our results.

## Figures and Tables

**Figure 1 cancers-16-00870-f001:**
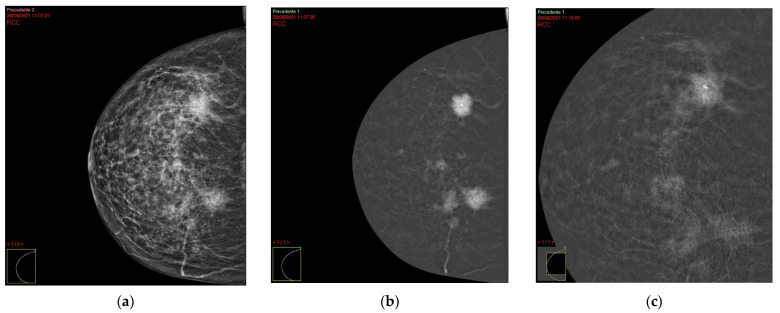
A 56-year-old woman with invasive ductal carcinoma not otherwise specified of histological grade 1, a lower Her 2 expression and absence of nodes metastases. CC projection in shows a multifocal mass with spiculated margins (arrow) (**a**,**b**) with wash out in the late phase (**c**).

**Table 1 cancers-16-00870-t001:** CEM presentation of breast tumors.

Group	N	%
Mass-like	102	56
Mass-Non-Mass Like	76	42
No Enhancement	4	2

**Table 2 cancers-16-00870-t002:** CEM features of mass-like lesions.

		N	%
Shape	Oval or Round	52	51
	Irregular	50	49
Margins	Circumscribed	18	18
	Non circumscribed	84	82
Dimensions	<2 cm	48	47
	≥2 cm	54	53
Contrast Enhancement	Homogeneous	28	27
	Heterogeneous	70	69
	Rim Enhancement	4	4
Late images	Wash-in	28	27
	Wash-out	74	73

**Table 3 cancers-16-00870-t003:** Immunohistochemical prognostic factors distribution.

		N	%
ER	neg	16	8
	pos	166	92
PR	neg	32	18
	pos	150	82
Ki67	neg	32	18
	pos	150	82
HER2	neg	140	77
	pos	42	23

**Table 4 cancers-16-00870-t004:** Histopathologic cancer distribution.

Histotype	N	%
Cis	0	0
CDI	68	37
CLI	12	7
CDI/CLI + Cis	102	56

**Table 5 cancers-16-00870-t005:** Correlation between CEM findings and classical prognostic factors.

CEM Findings	Tumor Size (cm)	Node Status	Histological Grade
**Type**	**<2 cm**	**+**	G1–G2
	**>2 cm**	**−**	G3
Mass	48 (27%)	26 (14.6%)	24 (13.5%)
	54 (30.3%)	76 (42.7%)	78 (43.8)
Non mass	38 (21%)	14 (7.9%)	16 (9%)
	38 (21%)	62 (34.8%)	60 (33.7%)
*p* value	0.406	0.175	0.419
**Shape of a mass**			
Regular	58 (32.6%)	14 (7.9%)	18 (10.1%)
	30 (16.9%)	74 (41.6%)	70 (39.3%)
Irregular	28 (15.7%)	26 (14.6%)	22 (12.4%)
	62 (34.8%)	64 (36%)	68(38.2%)
*p* value	<0.001	0.03	0.324
**Margins of a mass**			
Regular	22 (12.4%)	8 (4.5%)	10 (5.6%)
	2 (1.1%)	16 (9%)	14 (7.9%)
Spiculated	64 (36%)	32 (18%)	30 (16.9%)
	90 (50.6%)	122 (68.5%)	124 (69.7%)
*p* value	<0.001	0.135	0.019
**Internal enhancement pattern**			
Homogeneous	38 (21.3%)	6 (3.4%)	10 (5.6%)
	12 (6.7%)	44 (24.7%)	40 (22.5%)
Heterogeneous	48 (27%)	34 (19.1%)	30 (16.9%)
	76 (42.7%)	90 (50.6%)	94 (52.8%)
Rim	0	4 (2.2%)	0
	4 (2.2%)	0	4 (2.2%)
*p* value	0.001	<0.05	0.462
**Late phase**			
Wash in	18 (10.1%)	4 (2.2%)	10 (5.6%)
	24 (13.5%)	38 (21.3%)	32 (18%)
Wash out	68 (38.2%)	36 (20.2%)	30 (16.9%)
	68 (38.2%)	100 (56.2%)	106 (59.6%)
*p* value	0.256	0.014	0.481

**Table 6 cancers-16-00870-t006:** Correlation between CEM findings and immunohistochemical prognostic factors.

CEM Findings	ER	PR	Her 2	Ki-67
**Type**	**+**	**+**	**+**	**+**
	**−**	**−**	**−**	**−**
Mass	92 (51.7%)	84 (47.7%)	24 (13.5%)	86 (48.3%)
	10 (5.6%)	16 (9.1%)	78 (43.8%)	16 (9%)
Non mass	72 (40.4%)	62 (35.2%)	18 (10%)	62 (34.8%)
	4 (2.2%)	14 (8%)	58 (32.6%)	14 (8%)
*p* value	0.204	0.411	0.559	0.388
**Shape of a mass**				
Regular	86 (48.3%)	74 (42%)	20 (11.2%)	80 (45%)
	2 (1.1%)	12 (6.8%)	68 (38)	8 (4.5%)
Irregular	78 (43.8%)	72 (41%)	22 (12.4%)	68 (38.2%)
	12 (6.7%)	18 (10.2%)	68 (38%)	22 (12.4%)
*p* value	0.006	0.194	0.463	0.005
**Margins of a mass**				
Regular	24 (13.5%)	22 (12.5%)	12 (6.7%)	22 (12.4%)
	0	0	12 (6.7%)	2 (1%)
Spiculated	140 (78.7%)	124 (70.5%)	30 (16.9%)	126 (71%)
	14 (7.9%)	30 (17%)	124 (69.7%)	28 (15.8%)
*p* value	0.121	0.012	0.002	0.185
**Internal enhancement pattern**				
Homogeneous	50 (28%)	48 (27.3%)	10 (5.6%)	46 (25.8%)
	0	0	40 (22.5%)	4 (2.2%)
Heterogeneous	112 (63%)	96 (54.5%)	32 (18%)	98 (55%)
	12 (6.7%)	28 (16%)	92 (51.7%)	26 (14.6%)
Rim	2 (1%)	2 (1%)	0	4 (2.2%)
	2 (1%)	2 (1%)	4 (2.2%)	0
*p* value	<0.001	<0.001	0.381	0.04
**Late phase**				
Wash in	34 (19%)	32 (18%)	6 (3.4%)	36 (20.2%)
	8 (4.5%)	10 (5.6%)	36 (20.2%)	6 (3.4%)
Wash out	130 (73%)	114 (64%)	36 (20.2%)	112 (63%)
	6 (3.4%)	20 (11.2%)	100 (56.2%)	24 (13.5%)
*p* value	0.005	0.298	0.07	0.403

**Table 7 cancers-16-00870-t007:** Logistic regression analysis with likelihood-ratio covariate selection method for histologic grade, nodes involvement, ER and Pgr expression, Ki-67 status and HER expression.

Variables	β ± sE, β	*p*	OR (95% CI)
For Histologic grade			
- Spiculated margins	−1.568 ± 0.610	0.010	0.208 (0.063–0.688)
- Dimensions	−1.087 ± 0.453	0.016	0.337 (0.139–0.819)
For nodes involvement			
- Spiculated margins	1.556 ± 0.681	0.022	4.742 (1.249–18.004)
- Shape lesion	−1.135 ± 0.547	0.038	0.322 (0.110–0.939)
For ER			
- Wash-out	−1.492 ± 0.644	0.021	0.225 (0.064–0.795)
For PgR			
- Shape lesion	−1.461 ± 0.514	0.004	0.232 (0.085–0.685)
- Wash-out	−1.256 ± 0.554	0.023	0.285 (0.096–0.842)
For Ki-67			
- Shape lesion	1.399 ± 0.647	0.031	4.049 (1.139–14.399)
- Dimensions	−1.632 ± 0.501	0.001	0.195 (0.073–0.522)
For Her-2			
- Spiculated margins	2.466 ± 0.668	<0.001	11.773 (3.179–43.598)

## Data Availability

Data are unavailable because of the privacy.

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
