# Peer review of "The Correlation between Morpho-Dynamic Contrast-Enhanced Mammography (CEM) Features and Prognostic Factors in Breast Cancer: A Single-Center Retrospective Analysis"

_cancers, 2024, doi:10.3390/cancers16050870_

Round 1

Reviewer 1 Report

Comments and Suggestions for Authors

Dear Authors,

congratulations for this manuscript covering new imaging techniques like CEM drawing conclusions from the appearance of lesions in contrast-enhancement and categorizing them to tumor biologies. With this concept, patients can already have a close estimate of their diagnosis, even before  biopsy. 

I have some suggestions to make regarding minor language editing and also to the threshold of Ki 67. You chose "14 %" as the threshold to positivity. This is not in line with international standards, as Ki 67 is a continuous variable (Denkert C et. al ). Due to St.Gallen International Breast Cancer Consensus Conference it should be at 25 %. Please explain why you chose this threshold!

With best regards, 

PK

Comments on the Quality of English Language

Please change the spelling errors:

line 69: blank space before „prognostic"

line 116: blank space after „pattern“

line 164: internal instead of „Internal“

line 210: ER instead „Er“

line 211/212: Trastuzumab instead of „Herceptin“

Author Response

  1. .You chose "14 %" as the threshold to positivity. This is not in line with international standards, as Ki 67 is a continuous variable (Denkert C et. al ). Due to St.Gallen International Breast Cancer Consensus Conference it should be at 25 %. Please explain why you chose this threshold”.

Thank you for your suggestion. The ki-67 has a low reproducibility, as the literature reports, so we usually consider low proliferation tumors those with a ki-67 index <14%,  in accordance with both oncologists and pathologists; so we set up the cut off of Ki-67 at that level.

  1. Please change the spelling errors:”

“ line 69: blank space before „prognostic" ” Thank you for this suggestion. We modified it in the text.

line 116: blank space after „pattern“ Thank you for this suggestion. We modified it in the text.

line 164: internal instead of „Internal“ Thank you for this suggestion. We modified it in the text.

line 210: ER instead „Er“ Thank you for this suggestion. We modified it in the text.

line 211/212: Trastuzumab instead of „Herceptin“  Thank you for this suggestion. We modified it in the text.

Reviewer 2 Report

Comments and Suggestions for Authors

The paper entitled „Correlation between Morpho-Dynamic Contrast-Enhanced Mammography (CEM) Features and Prognostic Factors in  Breast Cancer: a Single-Center Retrospective Analysis” by Piccolo et al. needs the following improvements:

1. Please insert a pipeline or flowchart of your paper and a paragraph that describes the content of each section, as well as your main contributions. 

2. Please insert a section "Related work", with state of art methods to which this study refers.

3. The study is related to Contrast-Enhanced Mammography in section 2.2, please add the informatics and mathematic approaches for CEM.

4. In the section „3.1. Imaging analysis” please add the sample images with contrast enhanced.

5. In the section „3.3. Statistical analysis” the author used p-value, instead, this metric is not explained, and please use correct its value: e.g. line 198 (p=0,02), also in Table 5 (4(30,3%)), please check the entire manuscript.

6. The authors used „Logistic regression” but it is not described, in table 7 the value should contain a dot, not a comma. 

7. The section „Discussion”, should contain a table with „A comparison table of the state-of-the-art approaches” for inspiration you can see the paper „A Robust Machine Learning Model for Diabetic Retinopathy Classification”.

8. The limitations or drawbacks of this paper are not described. 

9. The Conclusion should contain future research. 

Author Response

Please find in attachment the answers to reviewer's comments.

After reviewing the whole manuscript, I kindly ask  to evaluate if the suggestions: 1)a "pipeline of the paper and a paragraph that describes the content of each section", which is inserted in the answers to reviewer but not yet in the manuscript; and 2)the informatics and mathematic approaches for CEM (added in the manuscript)are  necessary to make the paper more interesting or clear. 

  1. Please insert a pipeline or flowchart of your paper and a paragraph that describes the content of each section, as well as your main contributions. 

Thank you for your suggestions. We added in the text two tables describing lesions presentation on CEM. Moreover, a paragraph which describes the content of each section has been added below. We already submitted the author contribution during the first submission to the Editorial Office. Please refer to it for any issue.

This manuscript aims at describing our experience and results about the appearance of breast cancer at Contrast Enhanced Mammography (CEM). Specifically, in the Introduction we described the state of the art about the imaging techniques performed in the detection and characterization of breast cancer until now. To date, Magnetic Resonance Imaging is the gold standard technique to provide functional information about tumor neoangiogenesis, improving the detection and characterization of malignant lesion, but this is hindered by some drawbacks, such as patient’s claustrophobia, high costs, long acquisition times and limited availability. In the recent years CEM has shown diagnostic performance comparable to MRI in cancer detection and characterization.

In the Materials and Methods paragraph we described: 1) the inclusion and exclusion criteria; 2) CEM technique; 3) The image evaluation by applying the ACR lexicon; 4) The histological examination; and 5) The statistical analysis.

The results paragraph describes the imaging results analysis, in particular lesions appearance (mass-like, mass like and non- mass like and no enhancement), the features of mass lesions, paying particular attention at the late phase. Then we described the results of histo-pathological analysis, highlighting the values of the immunohistochemical prognostic factors and the histopathological results.

The statistical analysis was performed to evaluate the correlation between the morphological features and the prognostic factors, identified correlations between spiculated margins and irregular shape, and prognostic factors like tumor grade and molecular bio-markers.

In the Discussion section a brief summary about the existing literature on prognostic factor of breast cancer is given, as well as the role of MRI in this field. Then we described our results in comparison to others, which were performed by MR mainly, and highlighted the limits of our study. In the conclusions section we summarized briefly our results

  1. Please insert a section "Related work", with state of art methods to which this study refers.

Thank you for your suggestion. The state of the art the manuscript refers to is described in the introduction, with proper references.

  1. The study is related to Contrast-Enhanced Mammography in section 2.2, please add the informatics and mathematic approaches for CEM. Thank you for your suggestion. We added it.

  1. In the section „3.1. Imaging analysis” please add the sample images with contrast enhanced. Thank you very much for your suggestion. We submitted a case of breast cancer recognized on CEM in a separated file during the first submission. Please refer to it.

  1. In the section „3.3. Statistical analysis” the author used p-value, instead, this metric is not explained, and please use correct its value: e.g. line 198 (p=0,02), also in Table 5 (4(30,3%)), please check the entire manuscript.Thank you for your suggestion. We checked the manuscript carefully.
  2. The authors used „Logistic regression” but it is not described, in table 7 the value should contain a dot, not a comma.  Thank you for your suggestion. A section described the statistical analysis is present as paragraph 2.5. Table 7 has been modified
  3. The limitations or drawbacks of this paper are not described. The limitations has been added.
  4. The Conclusion should contain future research. Thank you for your suggestion. We modified the section according to it.

Reviewer 3 Report

Comments and Suggestions for Authors

The study retrospectivelly investigated association of CEM with breast cancer histology, grade and molecular markers. The investigation is well-designed, results are clearly presented, limitations are written. I suggest disscusion to be improved in context of more comparison with similar recent studies.

Author Response

Reviewer 3

The study retrospectivelly investigated association of CEM with breast cancer histology, grade and molecular markers. The investigation is well-designed, results are clearly presented, limitations are written. I suggest disscusion to be improved in context of more comparison with similar recent studies.

Thank you for your suggestion. We improved discussion.

Reviewer 4 Report

Comments and Suggestions for Authors

The article is a well-written and clinically interesting contribution to the field of breast cancer diagnosis. 

The "Introduction" part of the paper is very informative and the reasons why the study was conducted are carefully explained in this part. 

The "Patients and methods" and "Results" parts of the study are also very clearly described and presented. 

In the "Discussion" section of the study, the authors compared and discussed the topic in detail with similar work in this area. The results were also reported and compared with the results of other studies published in this field. 

However, I have few comments:

- Some abbreviations such as ER, PgR, HER2 etc. are not predefined

- The advantages of breast MRI over mammography, such as lower radiation exposure for patients with less harmful contrast agents during the examination and the simultaneous assessment of axillary lymph node metastases during breast MRI, should be emphasized in the discussion part of the manuscript. 

Author Response

Reviewer 4

Some abbreviations such as ER, PgR, HER2 etc. are not predefined

Thank you for your suggestion. We have modified the text.

- The advantages of breast MRI over mammography, such as lower radiation exposure for patients with less harmful contrast agents during the examination and the simultaneous assessment of axillary lymph node metastases during breast MRI, should be emphasized in the discussion part of the manuscript. 

Thank you for your suggestion. We have modified the text.

Round 2

Reviewer 2 Report

Comments and Suggestions for Authors

The equations are uncounted and the symbols can not be read, please rewrite.

In the section “2.5. Statistical Analysis” please insert mathematical approaches of logistic regression, the coefficients are explained but the logistic function is missed.

Line 184: the frequency is noted as n and in tables 1 and 2 is N.

The results are not compared with state-of-art methods.

Author Response

1 The equations are uncounted and the symbols can not be read, please rewrite.

            Thank you for the suggestion, we have included some readable equations.

2 In the section “2.5. Statistical Analysis” please insert mathematical approaches of logistic regression, the coefficients are explained but the logistic function is missed.

            Thank you for your suggestion, we have expanded this section in the text.

3 Line 184: the frequency is noted as n and in tables 1 and 2 is N.

            Thank you for your suggestion, we have modified the text.

The results are not compared with state-of-art methods.

Thank you for your observation. CEM is a relatively new technique which evaluates tumor neoangiogenesis in a similar way as MR; that’s the reason why in literature the most of studies focused on the comparison between these two techniques in terms of sensitivity, specificity and diagnostic accuracy as well as cancer detection rate. To date, this is the first study which correlates morphologic and dynamic features of histologically proven breast cancer with the classic and molecular prognostic factors, with a particular emphasis on the late phase of the study, which has not been yet extensively analyzed in the existing literature. According to our experience, the late phase gives us some interesting information about the dynamic behaviour of some histotypes of  breast cancer, as our results showed, that’s why we recommend to perform it in all the cases of locoegional staging of breast cancer. Because of the lacking of studies about CEM and prognostic factors, we compared our results with those obtained with MR. We aim that other studies could be performed on this topic, in order to assess the diagnostic feasibility of this technique.

The new modified manuscript is attached.
